# Red Cusk-Eel (*Genypterus chilensis*) Gut Microbiota Description of Wild and Aquaculture Specimens

**DOI:** 10.3390/microorganisms10010105

**Published:** 2022-01-04

**Authors:** Jaime Romero, Osmán Díaz, Claudio D. Miranda, Rodrigo Rojas

**Affiliations:** 1Instituto de Nutrición y Tecnología de los Alimentos (INTA), Universidad de Chile, El Líbano 5524, Macul, Santiago 783090, Chile; osman.diaz.r@gmail.com; 2Laboratorio de Patobiología Acuática, Departamento de Acuicultura, Universidad Católica del Norte, Larrondo 1281, Coquimbo 1780000, Chile; cdmirand@ucn.cl (C.D.M.); rrojas@ucn.cl (R.R.)

**Keywords:** microbiota, microbiome, *Genypterus chilensis*, *Tenericutes*, Ophidiidae, fish, aquaculture, next-generation sequencing

## Abstract

Chile has promoted the diversification of aquaculture and red cusk-eel (*Genypterus chilensis*) is one of the prioritized species. However, many aspects of the biology of the species are unknown or have little information available. These include intestinal microbiota, an element that may play an important role in the nutrition and defense of cultured animals for meat production. This study compares the microbiota composition of the intestinal contents of wild and aquaculture fish to explore the microbial communities present and their potential contribution to the host. DNA was extracted from the intestinal content samples and the V4 region of the 16S rRNA gene was amplified and sequenced using the Ion Torrent platform. After the examination of the sequences, strong differences were found in the composition at the level of phylum, being *Firmicutes* and *Tenericutes* the most abundant in aquaculture and wild condition, respectively. At the genus level, the *Vagococcus* (54%) and *Mycoplasma* (97%) were the most prevalent in the microbial community of aquaculture and wild condition, respectively. The evaluation of predicted metabolic pathways in these metagenomes showed that in wild condition there is an important presence of lipid metabolism belonging to the unsaturated fatty acid synthesis. In the aquaculture condition, the metabolism of terpenoids and polyketides were relevant. To our knowledge, this is the first study to characterize and compare the intestinal microbiota of red cusk-eel (*Genypterus chilensis*) of wild and aquaculture origin using high-throughput sequencing.

## 1. Introduction

Aquaculture is an important industry in some countries such as Chile, principally identified with salmon production, being the second producer of Atlantic salmon (*Salmo salar*) on the world [1]. However, different infectious diseases have recurrently affected Chilean salmon industry, which have leading to important economic losses [2], and consequently resulting in important sanitary and environmental problems. An important issue to be solved by the Chilean aquaculture is the low diversification of this activity, extremely concentrated in the salmonid culture, compromising its sustainability in the long-term. To develop a diverse aquaculture industry, not depending exclusively on the production of salmonid fish, the Chilean government has implemented the Chilean Aquaculture Diversification Program, mainly based on the development of aquaculture of various native fish species, such as the red cusk-eel (*Genypterus chilensis*), the one with good characteristics that make it a promising species to develop the technology for its cultivation [3].

Red cusk-eel (*Genypterus chilensis)* belongs to the Actinopterygii Clase, Telostei subclass, and Ophidiidae family, being a morphological characteristic of this family that the dorsal-fin rays usually are equal to or longer than opposing anal-fin rays [4]. Regarding *Genypterus chilensis*, it is a coastal bathydemersal endemic species, and it is distributed from Africa (18°25′ S) until Los Chonos Archipelago (47°70′ S) at depths ranging from 20 m to 150 m [5]. It has a carnivorous diet, being their common preys, crustaceans of small size, and fish [6,7]. The fishery of *Genypterus chilensis* is mainly limited to the non-industrial activity [4], with annual captures between 400 to 900 tons in the last years [1]. The *Genypterus* fishes have a high demand with an excellent economic profitability, nonetheless, many decades of over exploitation have put these species in a vulnerable situation.

The domestication of new fish species is a very complex challenge that involves the regulation of biological and non-biological factors to procure closed production life cycles [8]. The domestication key factors involve the study of the species biology, rearing technology, nutritional requirement, and control of the captive reproduction. Microbiota composition could be a very important factor in improving the culture, because the microbiota in fish has been shown to be important in a number of functions in the host, including prevention of gastrointestinal infection by niche competition or immunomodulatory effects, or provide a contribution to the nutritional status of the host [9,10,11,12]. The microbiota has been observed as an important factor affected by the captivity in aquaculture and its modulation could be critical in the future to control diseases in aquaculture [13]. In this context, the aim of this study is to evaluate possible differences in the microbiota associated to the intestinal contents of red cusk-eel of wild and aquaculture origin using next-generation sequencing (NGS) and to explore the potential importance of these differences to the host.

## 2. Materials and Methods

### 2.1. Sample Collection

Wild (*n* = 4) and aquaculture (*n* = 4) red cusk-eel *specimens* with an average 1.5 ± 0.5 kg of weight were collected. Animals considered in the study were previously analyzed for checking their sanitary status and diagnosed as healthy, corresponding to adult specimens, not exhibiting wounds, emaciation, deformities or any external symptom suggesting a non-healthy condition. A number of ten specimens of reared fish were then analyzed and from these only four specimens contained enough intestinal content to comply with the DNA extraction protocol. Then, for comparative purposes four specimens were considered to represent the wild population.

Aquaculture individuals (3-years-old) came from of the pisciculture Colorado Chile located in Coquimbo, Chile (30°00′45.4″ S 71°23′32.8″ W). Fish were euthanized with an overdose (200 mg/L) of tricaine methane sulfonate (MS222; Argent Chemical Laboratories, Redmond, WA, USA) and the samples of intestinal content were collected aseptically each individual by removing the digestive tract from the abdominal cavity and squeezing the hindguts, then freezing at −20 °C until DNA isolation. Wild specimens were speared by semi- autonomous diving scuba in Punta Teatinos (La Serena, Chile, 29°49′35.3″ S 71°16′57.1″ W) and kept in ice until processing. The samples of intestinal content were collected as described for reared fish. This study was conducted in accordance with the recommendations of the Guide for the Care and Use of Laboratory Animals of the National Institutes of Health.

### 2.2. DNA Extraction and Sequencing

DNA was extracted from intestinal content samples (0.25 g) using the MO BIO PowerFecal^®^ DNA Isolation Kit (MO BIO Laboratories, Carlsbad, CA, USA) following manufacturer’s protocol, including a previous treatment with 12 µL of lysozyme (MERCK) at 20 mg/mL at 37 °C for 60 min, then with 1.5 µL of proteinase K (Invitrogen) at 20 mg/mL at 37 °C for 60 min. DNA concentration was determined using the Qubit^®^ dsDNA BR Assay Kit (Life Technologies, Grand Island, NY, USA). The primers 515F (5′-GTGCCAGCMGCCGCGGTAA-3′) and 806R (5′-GGACTACHVGGGTWTCTAAT-3′) [14], were used for the amplification of the V4 hypervariable region. All PCR reactions contained 30 ng of DNA, 0.3 μL of each primer at 20 mM, 1.2 μL of dNTPs at 4 μM and 0.3 μL of GoTaq^®^ G2 Flexi DNA Polymerase (Promega Corporation, Madison, WI, USA). PCR conditions were 94 °C for 5 min, followed by 35 cycles of 94 °C for 30 s, 56 °C for 30 s and 68 °C for 45 s. DNA sequencing was performed in Plymouth University (Plymouth, United Kingdom) using the Ion Torrent Life Technologies/314 Chip instrument platform.

### 2.3. Sequences Analysis

Sequencing reads were processed using UPARSE [15] pipeline and analyzed with QIIME [14]: in brief, the reads were assessed in its quality with FASTQC software, then quality and length filtering was performed with USEARCH algorithm. A ‘bar code’ was added to the resulting sequences to obtain a single FASTA file. Subsequently, the reads were replicated, sorted by abundance, and singletons and chimeras were discarded. The reads were clustered into operational taxonomic units (OTUs) based on 97% identity using USEARCH. Using the Ribosomal Data Project (RDP) as the reference database with a confidence of 0.9, taxonomic information was assigned to each OTU using QIIME. Based on sequence frequency, representative sequences were determined for each OTU, and representative sequences were aligned with the PyNAST algorithm. FastTree was used to determine phylogenetic relationships, based on representative sequence alignment. Graphics for the relative abundance of the composition of intestinal microbiota were obtained by making use of the R environment package ‘ggplot2’ [16].

### 2.4. Diversity Indexes and Statistical Analysis

Alpha diversity indexes, including community diversity (Simpson and Shannon index), richness (Chao-1) and phylogeny-based metrics (PD Whole Tree), were calculated using QIIME as described by [13]. The Mann–Whitney test was used to test the differences in alpha diversity (Shannon Diversity index, Simpson index, richness and PD Whole Tree) between the wild and aquaculture cusk-eel using GraphPad Prism 6 (GraphPad Software, Inc., La Jolla, CA, USA). Statistical significance was considered for a *p*-value < 0.05. Beta diversity measurements were assessed through a principal coordinate analysis (PCoA) performed on the phylogenetic beta-diversity matrix, obtained by weighted UniFrac analysis, using QIIME. EMPeror was used to visualize the PCoA plots from weighted UniFrac metrics.

Linear discriminant analysis effect size (LefSe) [17] was used to assess the differential abundance of the bacterial components of specimens from wild and aquaculture conditions. LEfSe combines the standard tests for statistical significance (Kruskal–Wallis test and pairwise Wilcoxon test) with linear discriminate analysis (LDA) to estimate the effect size of each differentially abundant feature. The alpha value for the factorial Kruskal–Wallis test is 0.05, and the threshold for the logarithmic LDA score for discriminative features is 2.0.

### 2.5. Prediction of Molecular Functions Using PICRUSt

Prediction of metagenomes from each samples, was realized adopting a closed reference OTU-picking strategy based on the Greengenes database (v.13.5), with a 97% sequence similarity threshold using the “pick_closed_reference_otus.py” QIIME script. The resulting OTU table, in biome format, was then utilized to generate inferred metagenomic data using PICRUSt [18] v.1.1.0 with the default parameters. Initially, the abundance values of each OTU were normalized to their respective predicted 16S rRNA copy numbers. Then, the predicted functional pathways were annotated using the Kyoto Encyclopedia of Genes and Genomes (KEGG) database. The accuracy of the predictions of the metagenomes was assessed by computing the NSTI (Nearest Sequenced Taxon Index), which is an index that indicates the relationship of the microbes in a particular sample to the bacterial genomes in a database. The metabolic pathways were identified by employing HUMAnN2 (The HMP Unified Metabolic Analysis Network) with the default settings. The *t*-test was used to identify bacterial functional pathways that were differentially abundant in intestinal microbiota of wild and aquaculture cusk-eel. All *p*-values were corrected for an FDR of 0.05. FDR corrected *p*-values below 0.05 (FDR < 0.05) were considered significant.

### 2.6. Enzymatic Activity of Intestinal Content

Intestinal content enzyme activities were measured using the API^®^ ZYM System (bioMérieux, Marcy-l’Etoile, France) as described by [19] for fecal samples. Briefly, intestinal content samples (0.2 g of each) were resuspended in 1.5 mL sterile physiological saline solution (NaCl 0.85%) and were mixed gently and left to decant for 5 min. The API^®^ ZYM strips were inoculated with an aliquot of the supernatant and incubated at 30 °C for 4 h, according to the manufacturer’s instructions. Enzyme activity was graded from 1 to 5 (1 = 5 nM, 2 = 10 nM, 3 = 20 nM, 4 = 30 nM, 5 = 40 nM) based on color formation intensity according to the color chart provided by the manufacturer. In order, to graphically represent the results, the score was transformed into a percent value, where a score of 1 corresponds to 0% and 5 to 100%.

## 3. Results

### 3.1. Sequencing Depth

Intestinal contents were collected from wild (*n* = 4) and aquaculture (*n* = 4) *G. chilensis*. The microbiota composition was analyzed using barcoding sequencing of the V4 region of the 16S rRNA gene. After filtering, a total of 498,137 high-quality 16S rRNA gene amplicon sequences were obtained from the red cusk-eel analyzed, of which 212,463 reads come from the wild individuals and the other 285,674 reads from aquaculture individuals. These sequences have a 270 bp of length, and were distributed between 331 OTUs using QIIME based on 97% similarity.

### 3.2. Alpha and Beta Diversity Analyses

To characterize the bacterial diversity and richness in the gut of wild and aquaculture the alpha diversity indices were compared. Diversity values (Shannon and Simpson indexes), richness (Chao1) and phylogenetic diversity (PD Whole Tree) didn’t differ between the wild and aquaculture fish (Figure 1).

The beta diversity it’s represented in PCoA Plot based on the unweighted metric analysis (Figure 2) and showed that bacterial communities were grouped by origin (wild or aquaculture). The first three components explain a 79.27% of the variation. Furthermore, ANOSIM and PERMANOVA test (*p*-value < 0.05), confirmed a significant difference in the composition of the microbiota between wild an aquaculture fish (Table 1).

### 3.3. Microbiota Composition Differences between Wild and Aquaculture Cusk-Eel

Results of the taxonomic assignment are shown in Figure 3. The six dominant phyla: *Firmicutes*, *Tenericutes*, *Proteobacteria*, *Spirochaetes*, *Fusobacteria* and *Actinobacteria*, contributed up to the 98% of the total bacterial diversity. Three main phyla (*Firmicutes*, *Proteobacteria* and *Actinobacteria*) where consistently the most abundant in the aquaculture fish; summed together they accounted for more than 99%. On the other hand, *Tenericutes*, *Firmicutes* and *Proteobacteria* dominated the gut microbiota from wild samples, representing the 87.1%.

A relative higher abundance of the Phylum *Firmicutes* was observed across all aquaculture fish (90.3%) as compared to the wild samples (6.7%). In contrast, *Tenericutes* were more abundant in wild samples (60.4%) than in the wild fish (<1%). Likewise, *Proteobacteria* showed a relative abundance of 19.9% in wild individuals, whereas in aquaculture individuals had 6.26%.

At the genus level, important differences in relative abundances of each genus were observed between wild or aquacultured origin of the samples. In wild cusk-eel, the most predominant genera was *Mycoplasma* (59.3%), followed by *Vibrio* (8.8%), *Brevinema* (6.4%), *Alivibrio* (4.7%) and *Photobacteria* (4.2%). In aquaculture cusk-eel, *Vagococcus* (35.4%), *Facklamia* (22.0%), *Aerococcus* (10.6%), *Desemzia* (7.0%), *Photobacterium* (5.0%) and *Enterococcus* (4.8%) were the major bacterial genera in aquaculture intestinal content. *Mycoplasma* was the dominant genus is wild fish (59.3%) compared to aquaculture fish (<1%). In contrast, aquaculture fish have a notable proportion of *Vagococcus* (35.4%), *Faklamia* (22.0%), and *Aerococcus* (10.6%) which were more abundant compared to wild fish (<1% summed together).

### 3.4. Differential Bacterial Populations Associated to Each Condition

Differences in the relative abundance of bacterial taxa were analyzed using LEfSe, and the results are shown in Figure 4. The LEfSe analysis was used to determine the taxa that most likely explain the differences between wild and aquaculture samples. The analysis identified 35 bacterial taxa with significant differential relative abundances between the wild and aquaculture groups with LDA score >3.6. Sixteen of these bacterial taxa were higher in the wild fish and nineteen were higher in the aquaculture group (*p* < 0.005).

At phylum level, wild fish had higher abundances of *Tenericutes* and *Spirochaetes* as compared to aquaculture group. In contrast, the aquaculture samples were characterized by the presence of *Firmicutes* and *Actinobacteria*. In addition, these results confirm a significant enrichment of the genera *Mycoplasma*, *Brevinema*, *Psychrobacter*, *Cohnella* in wild cusk-eel, whereas *Carnonbacterium*, *Trichococcus*, *Pisciglobus*, *Desemzia*, *Facklamia* and *Vagococcus* are more abundant in aquaculture fish.

### 3.5. Functional Pathways with Significant Differences between Conditions

To investigate changes in the presumptive functions of the intestinal microbiota of red cusk-eel, HUMaN2 analysis was realized based on metagenomes prediction by PICRUSt. A total of 41 functional pathways showed significant differences (Figure 5; Appendix A); 18 of those pathways were more abundant in wild condition, including pathways related to lipid, cofactors and vitamins and carbohydrates metabolism. The other 23 pathways showed a higher abundance in aquaculture condition, including metabolism of terpenoids and polyketides, cofactors and vitamins and carbohydrates (Figure 5).

### 3.6. Enzymatic Activities Detected in Intestinal Contents

In aquaculture condition, enzymatic activities related to protein degradation were detected (Figure 6), such as Trypsin, Valine arylamidase and Leucine arylamidase. In wild condition, detected activities are related to carbohydrate degradation, such as α-Fucosidase, N-Acetyl-β-glucosaminidase, α-Glucosidase, β-Glucoronidase, β-Galactosidase and also Naphthol-AS-BI-Phosphohydrolase, Acid and Alkaline Phosphatase.

## 4. Discussion

Red cusk eel, *Genypterus chilensis* (Guichenot 1848), is a native Chilean species recognized as an iconic species in Chilean gastronomy due to its taste, cultural heritage (favorite dish of Chilean Nobel Prize Pablo Neruda) and nutritional value [20]. Overhunting and climate change may have contributed to a reduction in natural population size and captures [3]. For those reasons, red cusk eel, *G. chilensis* is projected as a candidate for the development of farming technology.

Despite the great importance of the studies on fish gut microbiota to improve welfare of fish and aquaculture practices, relative few studies on the effects of wild and aquaculture conditions on the microbiota have been done to the date. To our knowledge this is the first study using high-throughput sequencing for characterize and compare the intestinal microbiota between wild and reared *G. chilensis*.

Our results indicate that wild *G. chilensis* has a different microbiota compared to the aquaculture animals (Figure 2, Figure 4, Table 1). This indicates that the conditions in aquaculture systems lead to the establishment of different bacteria in the intestine as compared comparison to wild specimens, due to controlled parameters such as diet, population density and water quality. These results could be associated with differences in fish feeding depending on their origin. Significant changes in beta diversity have been observed in the *Salmo salar* intestinal microbiota when fed with different diets [21]. Schmidt et al. [22] reported similar observations suggesting that the *Salmo salar* microbiota was modulated by the diets under a RAS environment. These results are similar to other recent reports comparing wild and reared fish, such as *Paralichthys adspersus* [13], *Seriola lalandi* [23]; *Salmo salar* [24]; *Sparus aurata* [25]. Recently, Restivo et al. [26] described that the gut microbiome of wild fish changes with the transition to artificial environments (labs, tanks); they also showed that moving wild-caught fish into the lab altered alpha and beta-diversity. Alpha diversity was not different in our results and this is coincident to a previous report in fine flounder, but it contrasts observations in yellowtail (Appendix A).

In the present study, we showed that the dominant bacterial phylum present in the gut microbiota of the wild red cusk-eel was *Tenericutes* (60%), with *Mycoplasma* being the dominant genus. Previous studies have reported similar findings, where *Mycoplasma* was found to have the highest relative abundance in intestinal microbiota from wild rainbow trout and Atlantic salmon [24,27,28]. *Mycoplasma*, appear to be significantly affected by the growing conditions, because it was present in all wild fish with a higher abundance compared to the aquaculture fish that were detected at much lower levels. In contrast, *Firmicutes* was associated differentially to aquaculture condition of *G. chilensis*. Within this phylum, the most abundant genus was *Vagococcus*, which present evaluated probiotic effects of protection against vibriosis and effects in the innate immune system [29,30]. High abundance of *Firmicutes* is common among aquaculture fish, as previous works have reported *Firmicutes* as the most dominant bacterial phylum in carnivorous fish fed with commercial or experimental diets, such as salmonids [21,31,32]. It is important to notice that *Firmicutes* was the dominant phylum in reared fish (>60% in relative abundance, Appendix A) in other carnivorous Chilean species studied previously, fine flounder and yellowtail [13,23]. In contrast, those species showed the dominance of *Proteobacteria* in wild individuals, while wild red cusk-eel showed *Tenericutes* (Appendix A). This is novel in the field, because most reports of microbiota composition have indicated phyla *Firmicutes* and *Proteobacteria* as the higher relative abundance in several fish [33,34,35]. Recently, the term “comparative microbiomics” has arisen as a comparative approach between two or more species to determine the community structure, function and dynamics within its host, in order to gain perspective among species and taxa [36].

The difference in composition of the microbiota associated to *G. chilensis* from wild and aquaculture conditions may have an important effect on the host, in relation to the contribution of the functional pathways present in the components of the microbiota. Pathways associated with lipid, cofactors and vitamins metabolism were significantly associated with wild *G. chilensis* (Appendix A). Within lipid metabolism, linoleic acid and arachidonic acid metabolism, being linoleic acid the precursor of the arachidonic acid, and the latter is a precursor of eicosanoids such as prostaglandins and leukotrienes, which have roles in the immune system [37]. The metabolism of cofactors and vitamins include the metabolism of B vitamins, like biotin, that is important in the lipids and carbohydrates metabolism; the riboflavin, that plays a fundamental role in the energy metabolism; nicotinate and nicotinamide, precursors of nicotinamide adenine dinucleotide; and folate, essential in the synthesis of DNA and red blood cells; all of these have been proved to be synthetized by components of the microbiota [38,39]. On the other hand, in aquaculture *G. chilensis*, pathways of cofactors and vitamins metabolism and the metabolism of terpenoids and polyketides showed a higher and significant abundance. Within cofactors and vitamins metabolism, the retinol metabolism is related to vitamin A and has potent effects on the immune system when transformed into retinoic acid [40]. In the case of terpenoids and polyketides, pathways of interest included the biosynthesis of tetracycline and carotenoids, which correspond to an antibiotic and antioxidants, respectively. Terpenoids and polyketides present a wide range of activities and are important in the symbiotic interaction of a large number of sea and terrestrial species [41].

In the intestinal content samples from wild red cusk-eel, the enzymatic activities α-Glucosidase, β-Glucuronidase and β-Galactosidase are determined mainly by carbohydrate present in the diet [42]. The activities α-Fucosidase and N-Acetyl-β-glucosaminidase in the gastrointestinal tract have been related to host-microbiota interaction, affecting the microbial community composition [43], besides N-Acetyl-β-glucosaminidase, may be related to chitin biodegradation, an important component of crustaceans (Appendix A), which are the main source of food for the wild red cusk-eel that inhabits the rocky bottoms of the continental shelf at a depth of 20–150 m and use caves for sheltering and haunting [7,44,45]. It is interesting that glycolytic enzymes have also been reported in mycoplasmas isolated in terrestrial species such cows and chicken [46]. Furthermore, the secretome of *Mycoplasma capricolum* revealed the presence of acid and alkaline phosphatase [47]. Phosphatase activity, especially alkaline phosphatase, has an important role in digestion, absorption and nutrient transport [48]. For the aquaculture specimens, the enzymatic activities detected in intestinal content were trypsin, valine arylamidase and leucine arylamidase. All these correspond to peptidase activities, relevant in absorption of proteins and peptides. This is most likely due the artificial diet given to reared fish, which contains higher amounts of protein than that available in the wild fish diet that consists mostly of crustaceans [45,49].

The culture of the marine fish red cusk eel *Genypterus chilensis* is currently considered a priority for Chilean aquaculture but a problem associated with the cultivation of this species is the high mortality, up to 90%, in the stage of larvae and fry. An example of this main problem are recurrent episodes of vibriosis events, which have prompted the need for the use of antibiotics [50]. The use of these has been questioned given the selection of bacteria resistant to antimicrobials and the imbalance in the intestinal microbiota known as dysbiosis [51,52]. The use of probiotics is proposed as an alternative to overcome those problems associated with the use of antimicrobials. Previous studies have reported the use of autochthonous bacteria as an alternative to controlling bacterial diseases in fish culture [53,54,55]. As Nayak [12] and Mills et al. [56] described, compared to the benefits provided by bacteria isolated from other sources, beneficial bacteria isolated from a host could be more beneficial if administered again in the same species or similar species. In this sense, *Firmicutes* have a great interest for aquaculture, since it includes genera associated to protection against pathogens and immune system development, as lactic acid bacteria, which affect host resistance against pathogens and enhance the immune system response [35,55]. Appendix A showed the abundance in wild and reared red cusk eel, of possible candidates to further study as probiotics, such as *Lactococcus*, *Carnobacterium* and *Vagococcus*. These genera are easy to culture and there are previous report supporting their use as probiotics [29,30,57,58,59,60]. However, it must be noted that the results obtained in this study were obtained analyzing samples of whole intestinal contents, thus corresponding to the total gut microbiota. However, the distribution and percentages described could be different if different tracts of the fish gut were analyzed separately, as previously reported by other species, such as salmonids [60,61,62].

## 5. Conclusions

Our findings revealed deep differences between the microbiota composition of wild and aquaculture specimens, at phylum and genus level, due to farming conditions being immensely different from environment conditions where wild fish live. Presumptive metabolic pathways indicated that the microbiota contributes with many benefits to the host in both conditions (such as lipids metabolism, cofactors and biosynthesis of terpenoids and polyketides). The enzymatic activities in the intestinal content were different in both conditions, mostly due to diet composition, i.e., natural prey versus artificial diet. To our knowledge this is the first description of microbiota of *Genypterus chilensis* from environmental and culturable sources.

## Figures and Tables

**Figure 1 microorganisms-10-00105-f001:**
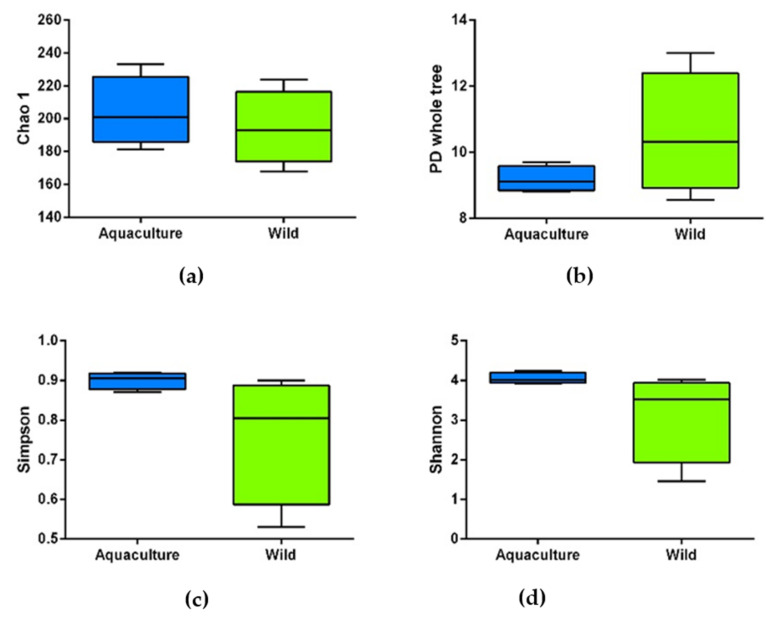
Comparison of alpha diversity indexes between wild and aquaculture red cusk-eel. Diversity in the intestinal bacterial community was measured using between wild (green) and aquaculture (blue). (**a**) Chao1; (**b**) PD Whole Tree; (**c**) Shannon index; (**d**) Simpson index.

**Figure 2 microorganisms-10-00105-f002:**
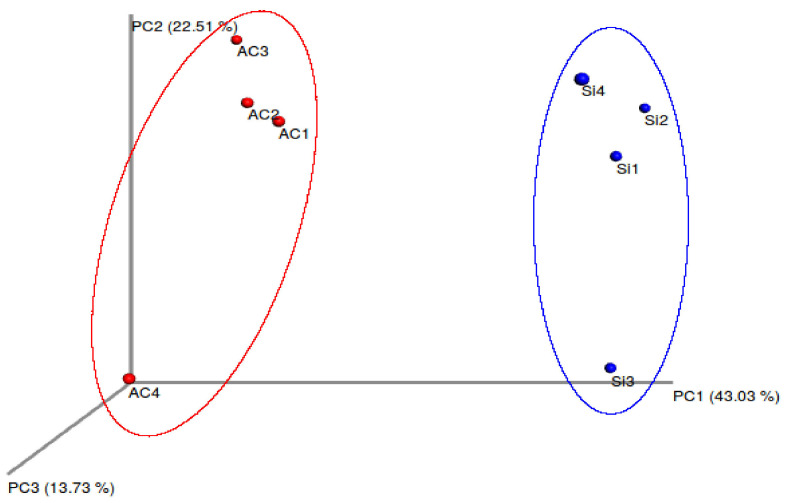
Beta diversity analysis of microbiota composition associated to *G. chilensis*. PCoA analysis of unweighted UniFrac distances of intestinal microbiota associated to different conditions (wild and aquaculture). Blue circles correspond to samples derived from wild fish (*n* = 4), and red circles correspond to samples from aquaculture fish (*n* = 4).

**Figure 3 microorganisms-10-00105-f003:**
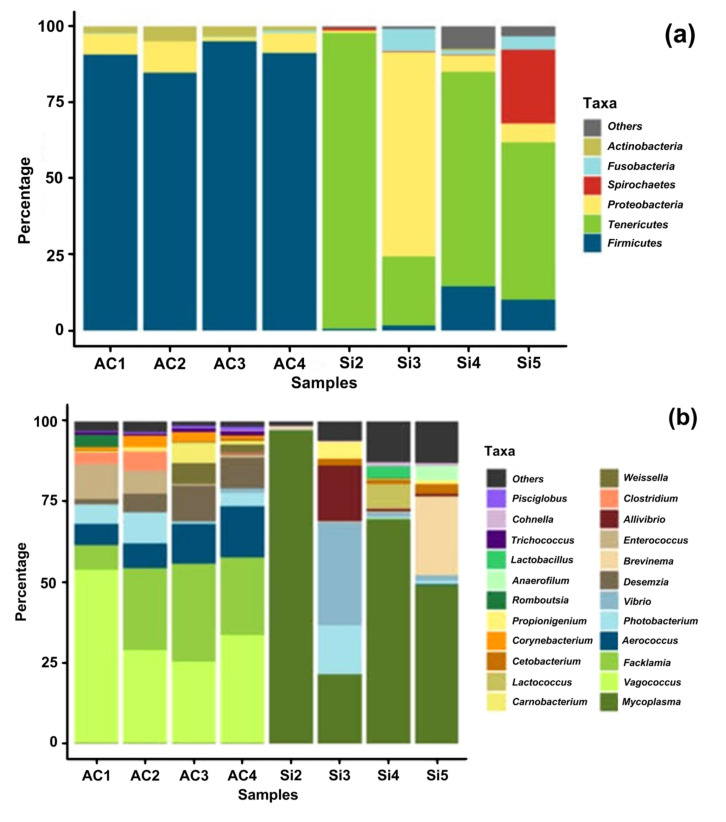
Relative abundances of OTU’s at two taxonomic levels in the intestinal microbiota from wild and aquaculture *G. chilensis***.** Histograms of relative abundances at (**a**) phylum and (**b**) genus level for all specimens (*n* = 8) A group represent to aquaculture samples, whereas W group to wild samples.

**Figure 4 microorganisms-10-00105-f004:**
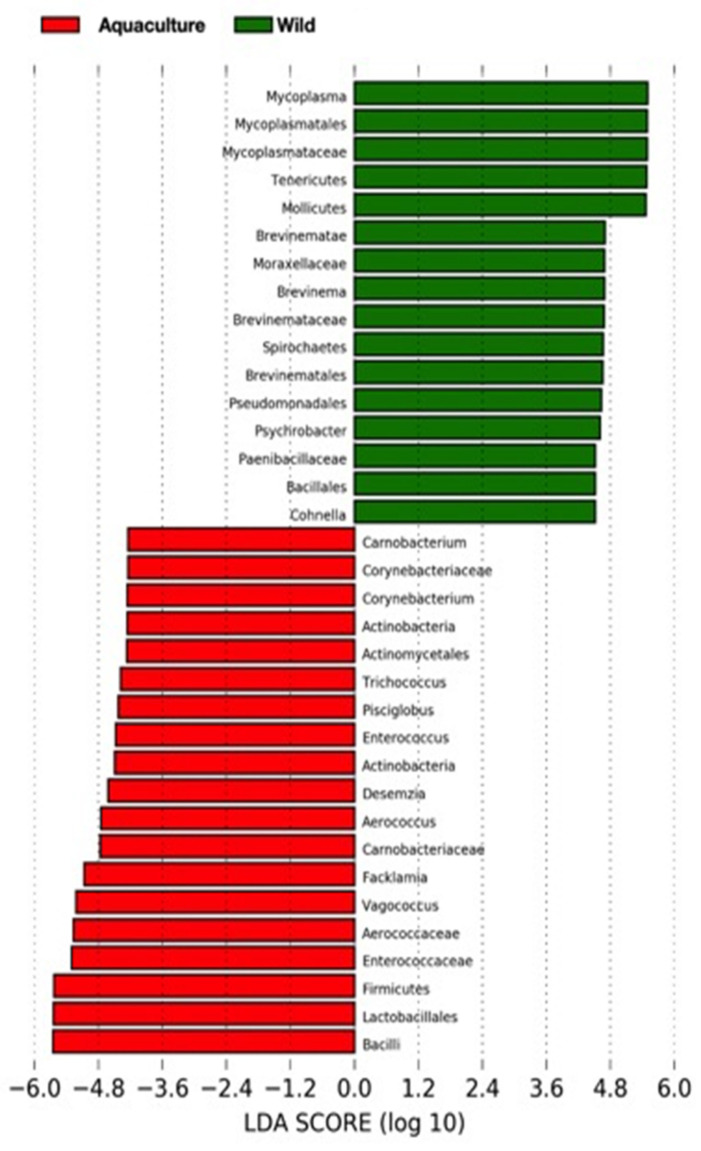
Characterization of gut microbiota in wild (green) and aquaculture (red) cusk-eel by LEfSe analysis. Histogram of the LDA scores (log_10_) of differentially abundant bacterial taxa, LDA scores showed the significant difference between the wild and aquaculture samples.

**Figure 5 microorganisms-10-00105-f005:**
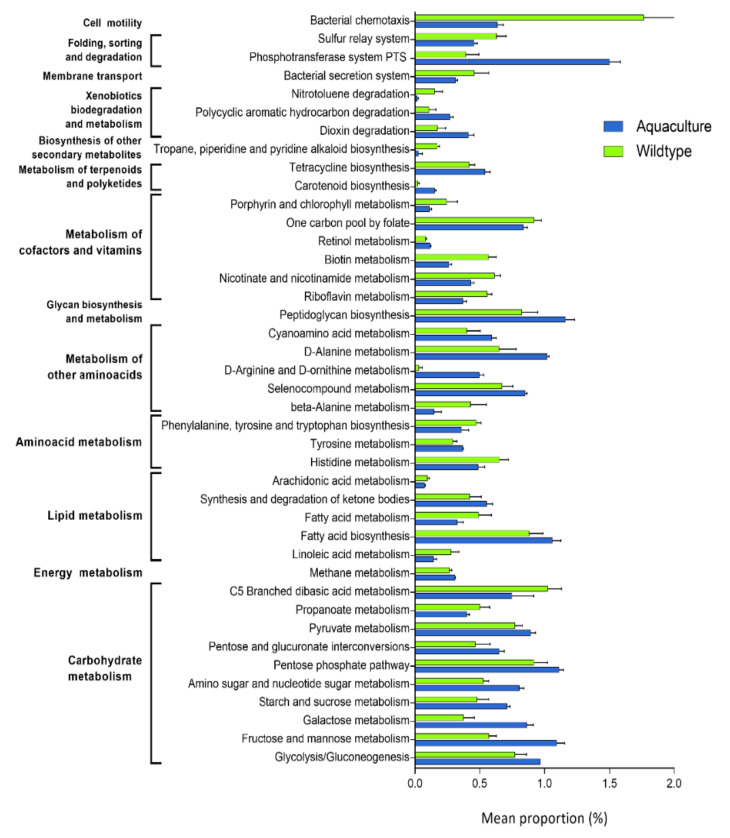
Presumptive functional pathways of the intestinal microbiota from wild and aquaculture red cusk-eel showing significant differences.

**Figure 6 microorganisms-10-00105-f006:**
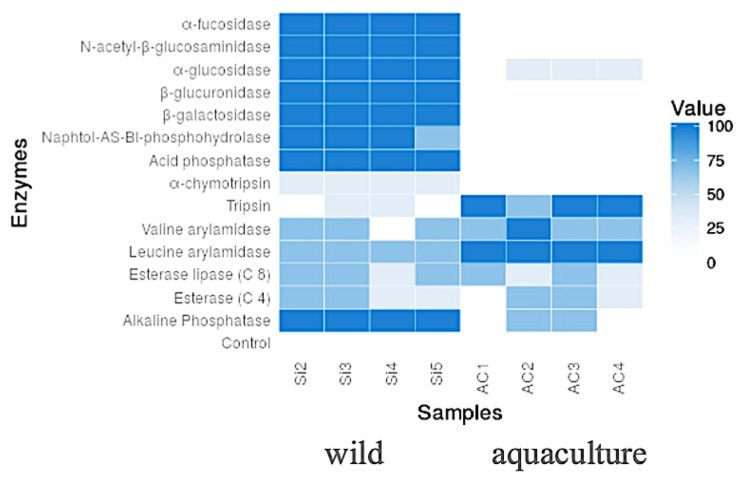
Heatmap showing enzymatic activity assessed by API^®^ ZYM system from wild and aquaculture fish.

**Table 1 microorganisms-10-00105-t001:** Beta diversity comparison between wild and aquaculture fish.

Matrix	ANOSIM (*p*-Value)	PERMANOVA (*p*-Value)
Weighted UniFrac	0.031	0.034
Unweighted UniFrac	0.025	0.022

*p*-value < 0.05.

## Data Availability

Data will be available on request.

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
