# Peer review of "Red Cusk-Eel (Genypterus chilensis) Gut Microbiota Description of Wild and Aquaculture Specimens"

_microorganisms, 2022, doi:10.3390/microorganisms10010105_

Round 1
Reviewer 1 Report
Dear Authors,
the manuscript “Red cusk-eel (Genypterus chilensis) gut microbiota description of wild and aquaculture specimens” is original and represents an important contribution to understand microbiota in wild and aquaculture red cusk-eel.
The study design is well described but, in my opinion, it lacks of a few important evaluations that could represent a bias for the results:
No evaluations of the sanitary status of the red cusk-eel are reported
The reason of the choice of four sample for wild and aquaculture is not explained; the number of samples is low (4+4) and it could not be representative of the two populations examined. Or is this number the result after a sanitary screening of the samples?
In the Methods section is reported the weight of the specimens but not the age; this data should be added. Considering the different diets of wild and aquaculture specimens there could be differences in the rate of growth and the weight could not represent the age of the samples (different ages could be related to a different microbiota).
In the beginning of the discussion Authors should report the concept expressed in the Introduction related to aquaculture diversification as tool against lost due to infectious diseases, that is sustainable strategy to have a resilient aquaculture. Moreover, always in this section, Authors should consider that results obtained were related to the total gut microbiota and the distribution and percentages described could be different if different tracts of the gut separately were analyzed.
A general check of the manuscript is required to remove a few typo errors:
Line 91: and the after pipeline to be removed
Line 152: G. chilensis to be written in italics
Figure 3: change Histogramas with histograms
Author Response
Answers to Reviewer 1
No evaluations of the sanitary status of the red cusk-eel are reported:
R: All sampled individuals considered in the study were previously diagnosed as healthy and the used procedure was properly described in the Materials and Methods section of the corrected version of the manuscript (Lines 71-74).
The reason of the choice of four sample for wild and aquaculture is not explained; the number of samples is low (4+4) and it could not be representative of the two populations examined. Or is this number the result after a sanitary screening of the samples?
R: The reviewer is right. The limited number is consequence of the reduced availability of specimens. The culture of this species is starting and Colorado have a limited production, mainly juveniles. A number of ten adult specimens were previously screened, but only 4 renderded enough intestinal content to obtain the amount of required DNA. Therefore, we analyzed the same number of wild fish to balance the study. We included this information in the Materials and Methods section (Lines 74-77).
In the Methods section is reported the weight of the specimens but not the age; this data should be added. Considering the different diets of wild and aquaculture specimens there could be differences in the rate of growth and the weight could not represent the age of the samples (different ages could be related to a different microbiota).
R: Only age of reared specimens are known, which were included in the corrected version of the manuscript (Line 78). We cannot be sure of the age of the wild fish. However, due to the similar weight we can assume a similar age.
In the beginning of the discussion Authors should report the concept expressed in the Introduction related to aquaculture diversification as tool against lost due to infectious diseases, that is sustainable strategy to have a resilient aquaculture.
R: We agree with the reviewer’s comment, and the referred paragraphs in the Introduction section were modified to avoid miss-understanding. The aquaculture diversification program in Chile has a different goal, as was clearly explained in this section of the corrected version of the manuscript (Lines 33-43).
Moreover, always in this section, Authors should consider that results obtained were related to the total gut microbiota and the distribution and percentages described could be different if different tracts of the gut separately were analyzed.
R: As was requested by the reviewer, we included in the Discussion section a proper analysis of this aspect of the study (Lines 367-371).
A general check of the manuscript is required to remove a few typo errors: Line 91: and the after pipeline to be removed
R: As was noted by the reviewer, “...pipeline and the and analyzed...” was changed to read “...pipeline and analyzed...” (Line 105).
Line 152: G. chilensis to be written in italics
R: As was noted by the reviewer, “G. chilensis” was written in italics (Line 166).
Figure 3: change Histogramas with histograms
R: As was noted by the reviewer, “Histogramas” was changed to read “Histograms” (Line 245).
Reviewer 2 Report
The manuscript aimed to describe and analyze the occurrence of hemagglutination caused by parasitism by Henneguya in Archosargus probatocephalus. Hematology is an important diagnostic tool, as it makes it possible to assess health conditions in different organisms. In fish, erythrocytes are cells sensitive to environmental variations, which can change their morphology and/or agglutinate, causing significant changes in the biological function of these cells. So far, few studies have described the natural occurrence of hemagglutination in fish and, for A. probatocephalus, it does not exist. What highlights the importance of the study.
Overall, the study is relevant and worthy of publication in the RPBV. The introduction is well written and conceptualizes the reader about the problem of the study. Materials and methods are adequate, results and discussion are appropriate, as is the conclusion.
Decision: minor revisions
Line 119: Inform the anesthetic dose.
Author Response
Answers to Reviewer 2
Line 119: Inform the anesthetic dose.
R: As was requested by the reviewer, anesthetic dose used in the study was included in the Materials and Methods section (Line 80).
Reviewer 3 Report
The study provides an interesting investigation of the gut microbiota, comparing samples from wild and cultivated red cusk-eel (Genypterus chilensis). The proposal is excellent, as it provides subsidies for the development of aquaculture.
Despite this, the study has many critical and negative points. Writing errors were observed, but the most critical can be seen in the small number of samples used in the study. In addition, the study has insufficient sampling sites, and a larger number of fish farms should be considered. Therefore, the article did not reach the publication level of this journal.
Author Response
Answer to Reviewer 3
The study provides an interesting investigation of the gut microbiota, comparing samples from wild and cultivated red cusk-eel (Genypterus chilensis). The proposal is excellent, as it provides subsidies for the development of aquaculture.
Despite this, the study has many critical and negative points. Writing errors were observed, but the most critical can be seen in the small number of samples used in the study. In addition, the study has insufficient sampling sites, and a larger number of fish farms should be considered. Therefore, the article did not reach the publication level of this journal.
R: Writing errors were corrected in this revised version.
R: Sampling sites: There is no other commercial farm than Colorado for the production of red cusk-eel in Chile.
R: Reduced number: The limited number is consequence of the reduced availability of specimens. The culture of this species is starting and Colorado has a limited production, mainly juveniles. A number of ten adult specimens were previously screened, but only 4 rendered enough intestinal content to obtain the amount of required DNA. Therefore, we analyzed the same number of wild fish to balance the study. We included this information in the Materials and Methods section (Lines 74-77).
Reviewer 4 Report
The manuscript with the title “Red cusk-eel (Genypterus chilensis) gut microbiota description of wild and aquaculture specimens” try to investigate the difference of microbiota composition between wild eel and aquaculture eel. This research topic has been widely researched in different fish species and got the nearly same conclusion that the microbiota of aquaculture fish is different from wild.
For this manuscript, the authors only catch 4 fish (wild or cultured) and the authors used intestinal content as the sample for the following analysis. This will lead to the miss conclusion because the microbiology in gut content nearly 100 percentage decided by feed but not fish.
The intestinal content enzymes activity analysis is not under the cover of the title and the authors did not give the connection between enzyme and microbiota.
Author Response
Answers to Reviewer 4
The manuscript with the title “Red cusk-eel (Genypterus chilensis) gut microbiota description of wild and aquaculture specimens” try to investigate the difference of microbiota composition between wild eel and aquaculture eel. This research topic has been widely researched in different fish species and got the nearly same conclusion that the microbiota of aquaculture fish is different from wild.
For this manuscript, the authors only catch 4 fish (wild or cultured) and the authors used intestinal content as the sample for the following analysis. This will lead to the miss conclusion because the microbiology in gut content nearly 100 percentage decided by feed but not fish.
R: Reduced number: The limited number is consequence of the reduced availability of specimens. The culture of this species is starting and Colorado has a limited production, mainly juveniles. A number of ten adult specimens were previously screened, but only 4 rendered enough intestinal content to obtain the amount of required DNA. Therefore, we analyzed the same number of wild fish to balance the study. We included this information in the Materials and Methods section (Lines 74-77).
R: gut content approach: We do not agree the reviewer assumption “in gut content nearly 100 percentage decided by feed but not fish”. Our reasons are: i) The use of the intestinal contents to assess the microbiota composition is an appropriate experimental approach, especially in fish of cold water habitats where the bacterial load is 2-3 log lower than in other animals such as homeotherms, i.e., 1x109 bacteria per gram in rainbow trout versus 1x1012 in human (Navarrete et al. 2010, doi:10.1111/j.1574-6941.2009.00769.x; Ilinskaya et al. 2017, doi: 10.3389/fmicb.2017.01666). ii) The gut wall may have even lower bacterial load, therefore, it is harder to get reliable microbial DNA. Furthermore, reduced of colonization of the gut wall, with the vast majority of the organisms residing
in lumen has been described in rainbow trout (Huber et. al. 2004, doi:10.1046/j.1365-2672.2003.02109.x). iii) The influence of the fish on the microbiota can be observed using intestinal content or feces. Contribution of host genetics in shaping the gut microbiota has been reported when the gut microbiota of different genetic families from breeding program has been assessed (Navarrete et al. 2012, doi:10.1371/journal.pone.0031335; Chapagain et al. 2020, doi:10.1186/s12864-020-07204-7).
The intestinal content enzymes activity analysis is not under the cover of the title and the authors did not give the connection between enzyme and microbiota.
R: Title: In our view, the title is appropriate because the core of the manuscript rely on the microbiota analysis. The enzymatic analysis is an only complement of the study; hence, it is not mentioned in the title.
R: Connection: As was requested by the reviewer, we included in the Discussion section a proper consideration of this aspect of the study (Lines 346-349).
Round 2
Reviewer 1 Report
Dear Authors,
thank you for revising the paper as requested.
Reviewer 3 Report
The present study requires reproducibility due to the low sampling effort. Make raw sequencing data readily available in a public database.
LINE 80: insert information about rearing conditions, production system, diet, environmental variables if available
LINE 86: insert information about wild conditions (oceanographic information, abiotic parameters)
LINE 276-281 - move to Introduction
LINE 286 - The present study compares few individuals from wild environments, with few individuals cultivated in a single fish farm. Please insert in the Discussion section that the interpretation of the data needs to be done carefully, and needs studies with higher sampling effort
Reviewer 4 Report
I still hold my oipnion and agree with the reviewer 3# . It is due to very small sample amount that limited the authors to check the colonized microbe. But It is the colonized microbe that conteract with the host.